# No Spurious Local Minima: on the Optimization Landscapes of Wide and Deep Neural Networks

## Abstract

Empirical studies suggest that wide neural networks are comparably easy to optimize, but mathematical support for this observation is scarce. In this paper, we analyze the optimization landscapes of deep learning with wide networks. We prove especially that constraint and unconstraint empirical-risk minimization over such networks has no spurious local minima. Hence, our theories substantiate the common belief that increasing network widths not only improves the expressiveness of deep-learning pipelines but also facilitates their optimizations.

## 1 Introduction

Deep learning depends on optimization problems that seem impossible to solve, and yet, deep-learning pipelines outperform their competitors in many applications. A common suspicion is that the optimizations are often easier than they appear to be. In particular, while most objective functions are nonconvex and, therefore, might have spurious local minima, recent findings suggest that optimizations are not hampered by spurious local minima as long as the neural networks are sufficiently wide. For example, Dauphin et al. (2014) suggest that saddle points, rather than local minima, are the main challenges for optimizations over wide networks; Goodfellow et al. (2014) give empirical evidence for stochastic-gradient descent to converge to a global minimum of the objective function of wide networks; Livni et al. (2014) show that the optimizations over some classes of wide networks can be reduced to a convex problem; Soudry & Carmon (2016) suggest that differentiable local minima of objective functions over wide networks are typically global minima; Nguyen & Hein (2018) indicate that critical points in wide networks are often global minima; and Allen-Zhu et al. (2019) and Du et al. (2019) suggest that stochastic-gradient descent typically converges to a global minimum for large networks.

These findings raise the question of whether common optimization landscapes over wide (but finite) neural networks have no spurious local minima altogether.

Progress in this direction has recently been made in Venturi et al. (2019) and then Lacotte & Pilanci (2020). Broadly speaking, we call a local minimum *spurious* if there is no nonincreasing path to a global minimum (see Section 2.2 for a formal definition). While the absence of spurious local minima does not preclude saddle points or suboptimal local minima in general, it means that one can move from every local minimum to a global minimum without increasing the objective function at any point—see Figure 1 for an illustration. Venturi et al. (2019) prove that there are no spurious local minima if the networks are sufficiently wide. Their theory has two main features that had not been established before: First, it holds for the entire landscapes—rather than for subsets of them. This feature is crucial: even randomized algorithms typically converge to sets of Lebesgue measure zero with probability one, that is, statements about "almost all" local minima are not necessarily meaningful. Second, their theory allows for arbitrary convex loss functions. This feature is important, for example, in view of the trends toward robust alternatives of the least-squares loss (Belagiannis et al., 2015; Jiang et al., 2018; Wang et al., 2016). On the other hand, their theory has three major limitations: it is restricted to polynomial activation, which is convenient mathematically but much less popular than ReLU activation, it disregards regularizers and constraints, which have become standard in deep learning and in machine learning at large (Hastie et al., 2015), and it restricts to

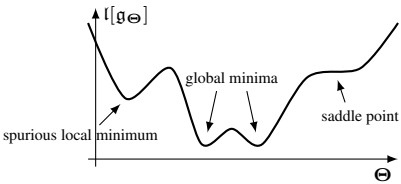

Figure 1: spurious local minimum of a hypothetical objective function

shallow networks, that is, networks with only one hidden layer, which contrasts the deep architectures that are used in practice (LeCun et al., 2015).

Lacotte & Pilanci (2020) made progress on two of these limitations: first, their theory caters to ReLU activation rather than polynomial activation; second, their theory allows for weight decay, which is a standard way to regularize estimators. However, their work is still restricted to one-hidden-layer networks. The interesting question is, therefore, whether such results can also be established for deep networks. And more generally, it would be highly desirable to have a theory for the absence of spurious local minima in a broad deep-learning framework.

In this paper, we establish such a theory. We prove that the optimization landscapes of empirical-risk minimizers over wide feedforward networks have no spurious local minima. Our theory combines the features of the two mentioned works, as it applies to the entire optimization landscapes, allows for a wide spectrum of loss functions and activation functions, and constraint and unconstraint estimation. Moreover, it generalizes these works, as it allows for multiple outputs and arbitrary depths. Additionally, our proof techniques are considerably different from the ones used before and, therefore, might be of independent interest.

**Guide to the paper**   Sections 2 and 5 are the basic parts of the paper: they contain our main result and a short discussion of its implications. Readers who are interested in the underpinning principles should also study Section 3, and readers who want additional insights on the proof techniques are referred to Section 4. The actual proofs are stated in the Appendix.

## 2   DEEP-LEARNING FRAMEWORK AND MAIN RESULT

In this section, we specify the deep-learning framework and state our main result. The framework includes a wide range of feedforward neural networks; in particular, it allows for arbitrarily many outputs and layers, a range of activation and loss functions, and constraint as well as unconstraint estimation. Our main result guarantees that if the networks are sufficiently wide, the objective function of the empirical-risk minimizer does not have any spurious local minima.

### 2.1   FEEDFORWARD NEURAL NETWORKS

We consider input data from a domain $\mathcal{D}_{\boldsymbol{x}} \subset \mathbb{R}^d$ and output data from a domain $\mathcal{D}_{\boldsymbol{y}} \subset \mathbb{R}^m$. Typical examples are regression data with $\mathcal{D}_{\boldsymbol{y}} = \mathbb{R}^m$ and classification data with $\mathcal{D}_{\boldsymbol{y}} = \{\pm 1\}^m$. We model the data with layered, feedforward neural networks, that is, we study sets of functions $\mathcal{G} := \{\mathfrak{g}_{\boldsymbol{\Theta}} : \mathcal{D}_{\boldsymbol{x}} \to \mathbb{R}^m : \boldsymbol{\Theta} \in \mathcal{M}\} \subset \overline{\mathcal{G}} := \{\mathfrak{g}_{\boldsymbol{\Theta}} : \mathcal{D}_{\boldsymbol{x}} \to \mathbb{R}^m : \boldsymbol{\Theta} \in \overline{\mathcal{M}}\}$ with

$$\mathfrak{g}_{\boldsymbol{\Theta}}[\boldsymbol{x}] := \Theta^l \mathfrak{f}^l \big[\Theta^{l-1} \cdots \mathfrak{f}^1[\Theta^0 \boldsymbol{x}]\big] \qquad \text{for } \boldsymbol{x} \in \mathcal{D}_{\boldsymbol{x}} \tag{1}$$

and

$$\mathcal{M} \subset \overline{\mathcal{M}} := \big\{\boldsymbol{\Theta} = (\Theta^l, \ldots, \Theta^0) : \Theta^j \in \mathbb{R}^{p^{j+1} \times p^j}\big\}.$$

The quantities $p^0 = d$ and $p^{l+1} = m$ are the input and output dimensions, respectively, $l$ the depth of the networks, and $\underline{w} := \min\{p^1, \ldots, p^l\}$ the minimal width of the networks. The functions $\mathfrak{f}^j : \mathbb{R}^{p^j} \to \mathbb{R}^{p^j}$ are called the activation functions. We assume that the activation functions are elementwise functions in the sense that $\mathfrak{f}^j[\boldsymbol{b}] = (\underline{\mathfrak{f}}^j[b_1], \ldots, \underline{\mathfrak{f}}^j[b_{p^j}])^\top$ for all $\boldsymbol{b} \in \mathbb{R}^{p^j}$, where $\underline{\mathfrak{f}}^j : \mathbb{R} \to \mathbb{R}$ is an arbitrary function. This allows for an unlimited variety in the type of activation, including ReLU $\underline{\mathfrak{f}}^j : b \mapsto \max\{0, b\}$, leaky ReLU $\underline{\mathfrak{f}}^j : b \mapsto \max\{0, b\} + \min\{0, cb\}$ for a fixed

$c \in (0, 1)$, polynomial $\mathfrak{f}^j : b \mapsto cb^k$ for fixed $c \in (0, \infty)$ and $k \in [1, \infty)$, and sigmoid activation $\mathfrak{f}^j : b \mapsto 1/(1 + e^{-b})$ as popular examples, and it allows for different activation functions in each layer.

We study the most common approaches to parameter estimation in this setting: constraint and unconstraint empirical-risk minimization. The loss function $\mathfrak{l} : \mathbb{R}^m \times \mathbb{R}^m \to \mathbb{R}$ is assumed convex in its first argument; this includes all standard loss functions, such as the least-squares loss $\mathfrak{l} : (\boldsymbol{a}, \boldsymbol{b}) \mapsto \|\boldsymbol{a} - \boldsymbol{b}\|_2^2$, the absolut-deviation loss $\mathfrak{l} : (\boldsymbol{a}, \boldsymbol{b}) \mapsto \|\boldsymbol{a} - \boldsymbol{b}\|_1$ (both typically used for regression), the logistic loss $\mathfrak{l} : (a, b) \mapsto -(1 + b) \log[1 + a] - (1 - b) \log[1 - a]$, the hinge loss $\mathfrak{l} : (a, b) \mapsto \max\{0, 1 - ab\}$ (both typically used for binary classification $\mathcal{D}_{\boldsymbol{y}} = \{\pm 1\}$), and so forth. The optimization domain is the set

$$\mathcal{M} := \big\{ \boldsymbol{\Theta} \in \overline{\mathcal{M}} : \mathfrak{r}[\boldsymbol{\Theta}] \leq 1 \big\}$$

for a constraint $\mathfrak{r} : \overline{\mathcal{M}} \to \mathbb{R}$. Given data $(\boldsymbol{x}_1, \boldsymbol{y}_1), \ldots, (\boldsymbol{x}_n, \boldsymbol{y}_n) \in \mathcal{D}_{\boldsymbol{x}} \times \mathcal{D}_{\boldsymbol{y}}$, the empirical-risk minimizers are then the networks $\mathfrak{g}_{\widehat{\boldsymbol{\Theta}}_{\mathrm{erm}}}$ with

$$\widehat{\boldsymbol{\Theta}}_{\mathrm{erm}} \in \arg\min_{\boldsymbol{\Theta} \in \mathcal{M}} \left\{ \sum_{i=1}^n \mathfrak{l}\big[\mathfrak{g}_{\boldsymbol{\Theta}}[\boldsymbol{x}_i], \boldsymbol{y}_i\big] \right\}. \tag{2}$$

It has been shown that constraints can facilitate the optimization as well as improve generalization—see Krizhevsky et al. (2012) and Livni et al. (2014), among others. For ease of presentation, we limit ourselves to the following class of constraints:

$$\mathfrak{r}[\boldsymbol{\Theta}] := \max\Big\{ a_{\mathfrak{r}} \max_{j \in \{1, \ldots, l\}} \|\Theta^j\|_1, b_{\mathfrak{r}} \|\Theta^0\|_q \Big\} \qquad \text{for all } \boldsymbol{\Theta} \in \overline{\mathcal{M}} \tag{3}$$

for fixed tuning parameters $a_{\mathfrak{r}}, b_{\mathfrak{r}} \in [0, \infty)$, a parameter $q \in (0, \infty]$, and $\|\cdot\|_q$ the usual row-wise $\ell_q$"-norm," that is, $\|\Theta^j\|_q := \max_k (\sum_i |(\Theta^j)_{ki}|^q)^{1/q}$ for $q \in (0, \infty)$ and $\|\Theta^j\|_\infty := \max_{ki} |(\Theta^j)_{ki}|$. This class of constraints includes the following four important cases:

- *Unconstraint estimation:* $a_{\mathfrak{r}} = b_{\mathfrak{r}} = 0$.

In other words, $\mathcal{M} = \overline{\mathcal{M}}$. Unconstraint estimation had been the predominant approach in the earlier days of deep learning and is still used today (Anthony & Bartlett, 1999).

- *Connection sparsity:* $q = 1$.

This constraint yields connection-sparse networks, which have received considerable attention recently (Barron & Klusowski, 2018; 2019; Kim et al., 2016; Taheri et al., 2020).

- *Strong sparsity:* $q < 1$.

Nonconvex constraints have been popular in statistics for many years (Fan & Li, 2001; Zhang, 2010), but our paper is probably the first one that includes such constraints in a theoretical analysis in deep learning.

- *Input constraints:* $a_{\mathfrak{r}} = 0$.

Some researchers have argued for applying certain constraints, such as node-sparsity, only to the input level (Feng & Simon, 2017). In general, while our proof techniques also apply to many other types of constraints, there are two main reasons for using the mentioned sparsity-inducing constraints to illustrate our results: First, sparsity has become very popular in deep learning, because it can lower the burden on memory and optimization as well as increase interpretability (Hebiri & Lederer, 2020). And second, the above examples allow us to demonstrate that the discussed features of wide networks do not depend on smooth and convex constraints such as weight decay.

Our theory can also be adjusted to the regularized versions of the empirical-risk minimizers, that is, for the networks indexed by any parameter in the set

$$\arg\min_{\boldsymbol{\Theta} \in \overline{\mathcal{M}}} \left\{ \sum_{i=1}^n \mathfrak{l}\big[\mathfrak{g}_{\boldsymbol{\Theta}}[\boldsymbol{x}_i], \boldsymbol{y}_i\big] + \mathfrak{r}[\boldsymbol{\Theta}] \right\}.$$

The proofs are virtually the same as for the constraint versions; we omit the details for the sake of brevity.

One line of research develops statistical theories for constraint and unconstraint empirical-risk minimizers—see Bartlett & Mendelson (2002) and Lederer (2020), among others. As detailed above, empirical-risk minimizers are the networks whose parameters are global minima of the objective function

$$\boldsymbol{\Theta} \;\mapsto\; \mathfrak{l}[\mathfrak{g}_{\boldsymbol{\Theta}}] := \sum_{i=1}^{n} \mathfrak{l}\big[\mathfrak{g}_{\boldsymbol{\Theta}}[\boldsymbol{x}_i], \boldsymbol{y}_i\big] \tag{4}$$

over $\mathcal{M}$ for fixed data $(\boldsymbol{x}_1, \boldsymbol{y}_1), \ldots, (\boldsymbol{x}_n, \boldsymbol{y}_n)$. While the function $\mathfrak{g}_{\boldsymbol{\Theta}} \mapsto \mathfrak{l}[\mathfrak{g}_{\boldsymbol{\Theta}}]$ is convex by assumption, the objective function $\boldsymbol{\Theta} \mapsto \mathfrak{l}[\mathfrak{g}_{\boldsymbol{\Theta}}]$ is usually nonconvex. It is thus unclear, per se, whether deep-learning pipelines can be expected to yield global minima of the objective function and, therefore, whether the statistical theories are valid in practice. Our goal is, broadly speaking, to establish conditions under which global minimization of (4) can indeed be expected.

## 2.2 Absence of Spurious Local Minima

We now show that the objective function (4) has no spurious local minima if the networks are sufficiently wide. Recall that a parameter $\boldsymbol{\Theta} \in \mathcal{M}$ that satisfies

$$\mathfrak{l}[\mathfrak{g}_{\boldsymbol{\Theta}}] \leq \mathfrak{l}[\mathfrak{g}_{\boldsymbol{\Gamma}}] \qquad \text{for all } \boldsymbol{\Gamma} \in \mathcal{M} \text{ with } \|\boldsymbol{\Theta} - \boldsymbol{\Gamma}\| \leq c$$

for a constant $c \in (0, \infty)$ and a norm $\|\cdot\|$ on $\overline{\mathcal{M}}$ is called a *local minimum* of the objective function (4). If the statement holds for every $c \in (0, \infty)$, the parameter $\boldsymbol{\Theta}$ is called a *global minimum*. Objective functions in deep learning have typically many local and global minima; an important question is whether there are "bad" local minima, that is, suboptimal local minima that are difficult to escape from. We formalize this notion as follows:

**Definition 1** (Spurious local minima). *Let $\boldsymbol{\Theta} \in \mathcal{M}$ be a local minimum of the objective function* (4). *If there is no continuous function $\mathfrak{h} : [0,1] \to \mathcal{M}$ that satisfies (i) $\mathfrak{h}[0] = \boldsymbol{\Theta}$ and $\mathfrak{h}[1] = \boldsymbol{\Gamma}$ for a global minimum $\boldsymbol{\Gamma} \in \mathcal{M}$ of the objective function* (4) *and (ii) $t \mapsto \mathfrak{l}[\mathfrak{g}_{\mathfrak{h}[t]}]$ is nonincreasing, we call the parameter $\boldsymbol{\Theta}$ a* spurious local minimum.

See again Figure 1 for an illustration.

The following theorem is our main result:

**Theorem 1** (Absence of spurious local minima). *Consider the setup of Section 2.1. If $\underline{w} \geq 2m(n+1)^l$, the objective function* (4) *has no spurious local minima.*

In other words, empirical-risk minimization over sufficiently wide networks does not involve spurious local minima. Hence, as long as there are means to circumvent saddle points (Dauphin et al., 2014), it is reasonable to expect that algorithms can find a global minimum and, therefore, that the known statistical theories for empirical-risk minimizers apply in practice.

The theorem applies very broadly. First, it includes all local minima rather than "many" or "almost all" local minima. This feature is important, because even randomized algorithms usually converge to a few, fixed points with high probability. Second, the framework allows for arbitrary convex loss functions. This feature caters, for example, to a current trend toward robust alternatives of the least-squares loss function (Barron, 2019; Lederer, 2020). Third, the framework includes ReLU activation. ReLU activation is nondifferentiable and, therefore, mathematically more challenging than, for example, linear and polynomial activation, but it has become the predominant type of activation in practice. Forth, the framework includes constraint as well as unconstraint estimation. Constraint estimation is particularly suitable for wide networks, and for overparameterized networks more generally, because it can avoid overfitting and facilitate optimizations. Fifth, our statement holds for arbitrary output dimensions and depths. The latter is particularly important in view of the current trend toward deep architectures. In sum, our result is a sweeping proof of the fact that wide networks have no spurious local minima, and it sheds light on the optimization landscapes of deep learning more generally.

The bound on the network widths becomes $2m(n+1)^l = 2(n+1)$ in the case of a single output ($m = 1$) and a single hidden layer ($l = 1$), which coincides with the bounds that have been established for

shallow networks with one output and specific activation functions and estimators—see Lacotte & Pilanci (2020) and references therein. Thus, our theory applies extremly broadly and still gives the expected results in the simple cases. In fact, our proofs only require one layer to have a width of at least $2m(n+1)^l$, but instead of losing ourselves in technical details about the condition, we focus on the main message of Theorem 1: optimizations become easier with increasing widths.

## 3 UNDERLYING CONCEPTS

In this section, we introduce concepts that we use in our proofs and that might also be of interest more generally. We first formulate the notion of path equivalence, which yields a practical characterization of spurious local minima. We then formulate specific parameters that can act as mediators between path-equivalent parameters. The main reason why our proof techniques are quite different from what can be found in the literature is that we cater to deep networks and a range of activation functions.

### 3.1 PATH RELATIONS

The objective functions for optimizing neural networks are typically continuous but not convex or differentiable. In the following, we characterize the absence of spurious local minima in a way that suits these characteristics of neural networks. The key concept is formulated in the following definition.

**Definition 2** (Path relations). *Consider two parameters $\Theta, \Gamma \in \mathcal{M}$. If there is a continuous function $\mathfrak{h}_{\Theta,\Gamma} : [0,1] \to \mathcal{M}$ that satisfies $\mathfrak{h}_{\Theta,\Gamma}[0] = \Theta$, $\mathfrak{h}_{\Theta,\Gamma}[1] = \Gamma$, and $t \mapsto \mathfrak{l}[\mathfrak{g}_{\mathfrak{h}_{\Theta,\Gamma}[t]}]$ is constant, we say that $\Theta$ and $\Gamma$ are* path constant *and write $\Theta \leftrightarrow \Gamma$.*

*If there is a continuous function $\mathfrak{h}_{\Theta,\Gamma} : [0,1] \to \mathcal{M}$ that satisfies $\mathfrak{h}_{\Theta,\Gamma}[0] = \Theta$, $\mathfrak{h}_{\Theta,\Gamma}[1] = \Gamma$, and $t \mapsto \mathfrak{l}[\mathfrak{g}_{\mathfrak{h}_{\Theta,\Gamma}[t]}]$ is convex, we say that $\Theta$ and $\Gamma$ are* path convex *and write $\Theta \smile \Gamma$.*

*If there are parameters $\Theta', \Gamma' \in \mathcal{M}$ such that (i) $\Theta \leftrightarrow \Theta'$ and $\Gamma \leftrightarrow \Gamma'$ and (ii) $\Theta' \smile \Gamma'$, we say that $\Theta$ and $\Gamma$ are* path equivalent *and write $\Theta \leftrightsquigarrow \Gamma$.*

Path constantness means that two parameters are connected by a continuous path of parameters that is constant with respect to the loss; path convexity relaxes "constant" to "convex;" path equivalence allows for additional mediators. The three relations are ordered in the sense that $\Theta \leftrightarrow \Gamma \Rightarrow \Theta \smile \Gamma \Rightarrow \Theta \leftrightsquigarrow \Gamma$, and they satisfy a number of other basic properties.

**Lemma 1** (Basic properties). *It holds for all $\Theta, \Gamma, \Psi \in \mathcal{M}$ that*

1. *$\Theta \leftrightarrow \Theta$; $\Theta \smile \Theta$; and $\Theta \leftrightsquigarrow \Theta$ (reflexivity);*

2. *$\Theta \leftrightarrow \Gamma \Rightarrow \Gamma \leftrightarrow \Theta$; $\Theta \smile \Gamma \Rightarrow \Gamma \smile \Theta$; and $\Theta \leftrightsquigarrow \Gamma \Rightarrow \Gamma \leftrightsquigarrow \Theta$ (symmetry).*

3. *$\Theta \leftrightarrow \Gamma$ and $\Gamma \leftrightarrow \Psi \Rightarrow \Theta \leftrightarrow \Psi$ (transitivity).*

The proof is straightforward and, therefore, omitted. The lemma illustrates that the path relations equip the parameter space with solid mathematical structures.

We can finally use the above-stated concepts to characterize spurious local minima.

**Proposition 1** (Characterization of spurious local minima). *Assume that for all $\Theta \in \mathcal{M}$, there is a global minimum of the objective function (4), denoted by $\Gamma$, such that $\Theta \leftrightsquigarrow \Gamma$. Then, the objective function (4) has no spurious local minima.*

Hence, path equivalence of all parameters to a global minimum is a sufficient condition for the absence of spurious local minima. This statement is the main result of Section 3.1.

### 3.2 BLOCK PARAMETERS

The parameterization of neural networks is typically ambiguous: many different parameters yield the same network. We leverage this ambiguity to make the networks more tractable. The key concept is formulated in the following definition.

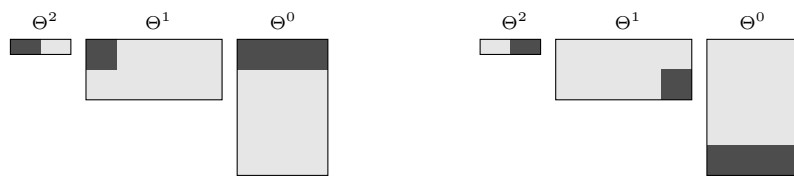

Figure 2: Illustration of the parameters of an $s$-upper block parameter (left) and an $s$-lower block parameter (right) for $l = 2$. The dark areas of the matrices can consist of arbitrary values; the light areas consist of zeros.

**Definition 3** (Block parameters). *Consider a number $s \in \{0, 1, \dots\}$ and a parameter $\boldsymbol{\Theta} \in \mathcal{M}$. If*

1. *$(\Theta^0)_{ji} = 0$ for all $j > s$;*

2. *$(\Theta^v)_{ij} = 0$ for all $v \in \{1, \dots, l-1\}$ and $i > s$ and for all $v \in \{1, \dots, l-1\}$ and $j > s$;*

3. *$(\Theta^l)_{ij} = 0$ for all $j > s$,*

*we call $\boldsymbol{\Theta}$ an $s$-upper-block parameter of depth $l$.*

*Similarly, if*

1. *$(\Theta^0)_{ji} = 0$ for all $j \leq p^1 - s$;*

2. *$(\Theta^v)_{ij} = 0$ for all $v \in \{1, \dots, l-1\}$ and $i \leq p^{v+1} - s$ and for all $v \in \{1, \dots, l-1\}$ and $j \leq p^v - s$;*

3. *$(\Theta^l)_{ij} = 0$ for all $j \leq p^l - s$,*

*we call $\boldsymbol{\Theta}$ an $s$-lower-block parameter of depth $l$. We denote the sets of the $s$-upper-block and $s$-lower-block parameters of depth $l$ by $\mathcal{U}_{s,l}$ and $\mathcal{L}_{s,l}$, respectively.*

Trivial examples are the $0$-block parameters $\mathcal{U}_0 = \mathcal{L}_0 = \{\mathbf{0} = (\mathbf{0}_{p^{l+1} \times p^l}, \dots, \mathbf{0}_{p^1 \times p^0})\}$ and the $s$-block parameters $\mathcal{U}_{s,l} = \mathcal{L}_{s,l} = \mathcal{M}$ for $s \geq \max\{p^1, \dots, p^l\}$. More generally, the block parameters consist of block matrices: see Figure 2. We show in the following that block parameters can be mediators in the sense of path equivalence.

We first show that every parameter is path constant to a block parameter.

**Proposition 2** (Path connections to block parameters). *For every $\boldsymbol{\Theta} \in \mathcal{M}$ and $s := m(n+1)^l$, there are $\overline{\boldsymbol{\Theta}}, \underline{\boldsymbol{\Theta}} \in \mathcal{M}$ with $\overline{\boldsymbol{\Theta}} \in \mathcal{U}_{s,l}$ and $\underline{\boldsymbol{\Theta}} \in \mathcal{L}_{s,l}$ such that $\boldsymbol{\Theta} \leftrightarrow \overline{\boldsymbol{\Theta}}$ and $\boldsymbol{\Theta} \leftrightarrow \underline{\boldsymbol{\Theta}}$.*

In particular, every parameter is path connected to both an upper-block parameter and a lower-block parameter. The interesting cases are wide networks: for fixed $s$, the wider the network, the more pronounced the block structure.

We then show that there is a connection between upper-block and lower-block parameters.

**Proposition 3** (Path connections among block parameters). *Consider two block parameters $\boldsymbol{\Theta} \in \mathcal{U}_{s,l}$ and $\boldsymbol{\Gamma} \in \mathcal{L}_{s,l}$. If $\underline{w} \geq 2s$, it holds that $\boldsymbol{\Theta} \leftrightsquigarrow \boldsymbol{\Gamma}$.*

Hence, every upper-block parameter is path connected to every lower-block parameter—as long as the minimal width of the networks is sufficiently large.

We finally combine Propositions 2 and 3.

**Corollary 1** (All parameters are path equivalent). *Consider two arbitrary parameters $\boldsymbol{\Theta}, \boldsymbol{\Gamma} \in \mathcal{M}$. If $\underline{w} \geq 2m(n+1)^l$, it holds that $\boldsymbol{\Theta} \leftrightsquigarrow \boldsymbol{\Gamma}$.*

See Figure 3 for an illustration. The corollary ensures that as long as the minimal width is sufficiently large, all networks are path equivalent. This result, therefore, connects directly to the characterization of spurious local minima in Proposition 1 of the previous section.

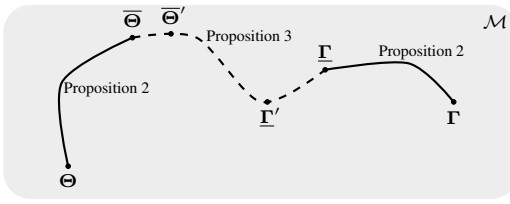 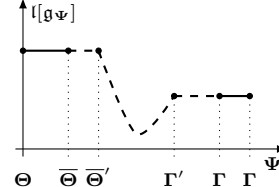

Figure 3: path-equivalence between two parameters $\Theta$ and $\Gamma$—see Corollary 1 and Definition 2

## 4 AUXILLIARY RESULTS

In this section, we state four auxilliary results.

### 4.1 TWO-LAYER NETWORKS

Here, we show that two-layer networks can be reparametrized such that they are indexed by block parameters. We first introduce the notation

$$\mathfrak{r}[M, q] := \|M\|_q = \max_{a \in \{1, \ldots, b\}} \left( \sum_{j=1}^{c} |M_{aj}|^q \right)^{1/q} \qquad \text{for all } M \in \mathbb{R}^{b \times c}, \ q \in (0, \infty)$$

and $\mathfrak{r}[M, \infty] := \|M\|_\infty = \max_{aj} |M_{aj}|$ for all $M \in \mathbb{R}^{b \times c}$. These functions are the building blocks of the constraint on Page 3. Next, given a permutation $\mathfrak{p} : \{1, \ldots, c\} \to \{1, \ldots, c\}$ and a matrix $M \in \mathbb{R}^{b \times c}$, we define the matrix $M_\mathfrak{p} \in \mathbb{R}^{b \times c}$ through $(M_\mathfrak{p})_{ij} := M_{i\mathfrak{p}[j]}$. Similarly, given a permutation $\mathfrak{p} : \{1, \ldots, b\} \to \{1, \ldots, b\}$ and a matrix $M \in \mathbb{R}^{b \times c}$, we define the matrix $M^\mathfrak{p} \in \mathbb{R}^{b \times c}$ through $(M^\mathfrak{p})_{ji} := M_{\mathfrak{p}[j]i}$. The result is then the following:

**Lemma 2** (Two-Layer networks). *Consider three matrices $A \in \mathbb{R}^{u \times v}$, $B \in \mathbb{R}^{v \times o}$, and $C \in \mathbb{R}^{o \times r}$, two constants $q_A \in (0, 1]$ and $q_B \in (0, \infty]$, and a function $\mathfrak{h} : \mathbb{R} \to \mathbb{R}$. With some abuse of notation, define $\mathfrak{h} : \mathbb{R}^{v \times r} \to \mathbb{R}^{v \times r}$ through $(\mathfrak{h}[M])_{ji} := \mathfrak{h}[M_{ji}]$ for all $M \in \mathbb{R}^{v \times r}$. Then, there are matrices $\overline{A} \in \mathbb{R}^{u \times v}$ and $\overline{B} \in \mathbb{R}^{v \times o}$ and a permutation $\mathfrak{p} : \{1, \ldots, v\} \to \{1, \ldots, v\}$ such that*

- $\overline{A}\mathfrak{h}[\overline{B}C] = A_\mathfrak{p}\mathfrak{h}[B^\mathfrak{p}C]$;

- $\mathfrak{r}[\overline{A}, q_A] \le \mathfrak{r}[A_\mathfrak{p}, q_A]$ *and* $\mathfrak{r}[\overline{B}, q_B] \le \mathfrak{r}[B^\mathfrak{p}, q_B]$;

- $\overline{A}_{ij} = 0$ *for* $j > u(r + 1)$; $\overline{B}_{ji} = 0$ *for* $j > u(r + 1)$ *and* $\overline{B}_{ji} = (B^\mathfrak{p})_{ji}$ *otherwise.*

*Similarly, there are matrices $\underline{A} \in \mathbb{R}^{u \times v}$ and $\underline{B} \in \mathbb{R}^{v \times o}$ and a permutation $\mathfrak{p} : \{1, \ldots, v\} \to \{1, \ldots, v\}$ such that*

- $\underline{A}\mathfrak{h}[\underline{B}C] = A_\mathfrak{p}\mathfrak{h}[B^\mathfrak{p}C]$;

- $\mathfrak{r}[\underline{A}, q_A] \le \mathfrak{r}[A_\mathfrak{p}, q_A]$ *and* $\mathfrak{r}[\underline{B}, q_B] \le \mathfrak{r}[B^\mathfrak{p}, q_B]$;

- $\underline{A}_{ij} = 0$ *for* $j \le v - u(r + 1)$; $\overline{B}_{ji} = 0$ *for* $j \le v - u(r + 1)$ *and* $\overline{B}_{ji} = (B^\mathfrak{p})_{ji}$ *otherwise.*

Hence, the parameter matrices of two-layer networks can be brought into the shapes illustrated in Figure 2. We apply this result repeatedly in the proof of Proposition 2.

### 4.2 SYMMETRY PROPERTY OF NEURAL NETWORKS

Next, we point out a symmetry in our setup for the neural networks.

**Lemma 3** (Symmetry property). *Consider permutations $\mathfrak{p}^j : \{1, \ldots, p^j\} \to \{1, \ldots, p^j\}$ for $j \in \{0, \ldots, l + 1\}$. Assume that $\mathfrak{p}^0$ and $\mathfrak{p}^{l+1}$ are the identity functions: $\mathfrak{p}^0[j] = \mathfrak{p}^{l+1}[j] = j$ for all $j$. The parameter $\Theta \in \mathcal{M}$ is a spurious local minimum of the objective function (4) if and only if $\Gamma \in \mathcal{M}$ defined through $(\Gamma^j)_{uv} := (\Theta^j)_{\mathfrak{p}^{j+1}[u]\mathfrak{p}^j[v]}$ for all $j \in \{0, \ldots, l\}$, $u \in \{1, \ldots, p^{j+1}\}$, and $v \in \{1, \ldots, p^j\}$ is a spurious local minimum of the objective function (4).*

The proof follows readily from our setup in Section 2.1 and, therefore, is omitted. The lemma illustrates that the parameterizations of neural networks are highly ambiguous. But in this case, the ambiguity is convenient, because it allows us to permute the rows and columns of the parameters to bring the parameters in shapes that are easy to manage.

### 4.3 PROPERTY OF CONVEX FUNCTIONS

We now establish a simple property of convex functions.

**Lemma 4** (Property of convex functions). *Consider a convex function $\mathfrak{h} : [0,1] \to \mathbb{R}$. If $\mathfrak{h}[0] > \mathfrak{h}[\bar{t}]$ for a $\bar{t} \in (0,1]$, there is a $c \in \arg\min_{t \in (0,1]}\{\mathfrak{h}[t]\}$ such that the function $\tilde{\mathfrak{h}} : [0,1] \to \mathbb{R}$ defined through $\tilde{\mathfrak{h}}[t] := \mathfrak{h}[ct]$ for all $t \in [0,1]$ is nonincreasing and $\tilde{\mathfrak{h}}[0] > \tilde{\mathfrak{h}}[1]$.*

This lemma connects the convexity from Definition 2 with the spurious local minima from Definition 1. We use this result in the proof of Proposition 1.

### 4.4 CARATHÉODORY-TYPE RESULT

Carathéodory's theorem goes back to Carathéodory (1911); see Boltyanski & Martini (2001); Fenchel (1929); Hanner & Rådström (1951) for related results. The following statement combines the classical theorem and the much more recent results Bastero et al. (1995, Theorem 1 and Lemma 1).

**Lemma 5** (Carathéodory-Type result). *Consider a number $q \in (0,1]$, vectors $\boldsymbol{z}_1, \ldots, \boldsymbol{z}_h \subset \mathbb{R}^r$, and the vectors' $q$-convex hull $\operatorname{conv}_q[\boldsymbol{z}_1, \ldots, \boldsymbol{z}_h] := \{\sum_{j=1}^h \tilde{t}_j \boldsymbol{z}_j : \tilde{\boldsymbol{t}} \in [0,1]^h, \|\tilde{\boldsymbol{t}}\|_q = 1\}$. Then, every vector $\boldsymbol{v} \in \operatorname{conv}_q[\boldsymbol{z}_1, \ldots, \boldsymbol{z}_h]$ can be written as $\boldsymbol{v} = \sum_{j=1}^h t_j \boldsymbol{z}_j$ with $\boldsymbol{t} \in [0,1]^h$, $\|\boldsymbol{t}\|_q \leq 1$, and $\#\{j \in \{1, \ldots, h\} : t_j \neq 0\} \leq r + 1$.*

The cardinality of a set $\mathcal{A}$ is denoted by $\#\{\mathcal{A}\}$ and the $\ell_q$-norm of a vector $\boldsymbol{v} \in \mathbb{R}^a$ by $\|\boldsymbol{v}\|_q := (\sum_{j=1}^a |v_j|^q)^{1/q}$. The lemma follows readily from the mentioned results; therefore, its proof is omitted. The lemma states that every vector in the $q$-convex hull is a $q$-convex combination of at most $r+1$ vectors from the set of vectors that generate the $q$-convex hull. (Note that for $q < 1$, our definition of the $q$-convex hull is more restrictive than the standard definition—cf. Bastero et al. (1995, Remark on Page 142)—but it leads to a concise statement and is sufficient for our purposes.)

## 5 DISCUSSION

Empirical evidence has long suggested that wide networks are comparatively easy to optimize. In this paper, we underpin these observations with rigorous theory: we prove that the optimization landscapes of empirical-risk minimization over wide networks have no spurious local minima.

Standard competitors to deep learning are "classical" high-dimensional estimators, such as the ridge and the lasso (Hoerl & Kennard, 1970; Tibshirani, 1996), that have been studied in statistics extensively (Zhuang & Lederer, 2018). A common argument for these estimators is that their objective functions are often convex or equipped with efficient algorithms for optimization (Bien et al., 2018; Friedman et al., 2010), but our results indicate that global optimization of the objective functions in deep learning can be feasible as well.

Our framework allows for arbitrary depths, for constraint as well as unconstraint estimation, essentially arbitrary activation, and for a very wide spectrum of loss functions and input and output data. This generality demonstrates that the absence of spurious local minima is not a feature of specific estimators, network functions, or data but, instead, a universal property of wide networks. Our theory, therefore, supports the use of wide networks in general—possibly together with regularization or constraints to avoid overfitting.

The main idea of most approaches in the field is to construct basis functions for the networks. In contrast, we formulate parametrizations that make the networks easy to work with. We could thus envision testing our new concepts beyond the presented application.

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

## A PROOFS

### A.1 PROOF OF THEOREM 1 AND AN EXTENSION

*Proof of Theorem 1.* The proof combines the main results of Sections 3.1 and 3.2. Let $\Theta \in \mathcal{M}$ be an arbitrary parameter and $\Gamma \in \mathcal{M}$ a global minimum of the objective function (4). In view of Proposition 1 in Section 3.1, we need to show that $\Theta \leftrightsquigarrow \Gamma$, and this follows directly from Corollary 1 in Section 3.2. $\qquad\square$

Figure 1 illustrates that a nonconvex objective function can have spurious local minima, but Theorem 1 proves that this is not the case for deep learning with wide networks. Figure 1 also illustrates that a nonconvex objective function can have "disconnected" global minima, but, in view of Corollary 1 and our results more generally, we can rule out this case as well. In other words, the set of global minima is "connected" in the sense that it is a set of path-constant parameters. The connectedness of the global minima is of minor importance in practice but an interesting topological property nevertheless.

### A.2 PROOF OF PROPOSITION 1

*Proof of Proposition 1.* The key idea is to exploit the properties of monotone functions and convex functions. Let $\Theta \in \mathcal{M}$ be an arbitrary parameter and $\Gamma \in \mathcal{M}$ a global minimum of the objective function (4) such that $\Theta \leftrightsquigarrow \Gamma$. Let $\Theta', \Gamma' \in \mathcal{M}$ and $\mathfrak{h}_{\Theta',\Gamma'} : [0,1] \to \mathcal{M}$ be as described in Definition 2. Note first that, since $\Gamma \leftrightarrow \Gamma'$ (see Definition 2), the parameter $\Gamma'$ is also a global minimum of the objective function. We then use 1. simple algebra, 2. the assumed convexity of the function $t \mapsto \mathfrak{l}[\mathfrak{g}_{\mathfrak{h}_{\Theta',\Gamma'}}]$ (see Definition 2 again), 3. the assumed endpoints of the function $\mathfrak{h}_{\Theta',\Gamma'}$ (see Definition 2 once more), 4. the fact that $\Gamma'$ is a global minimum of (4), and 5. the fact that $(1-t)+t=1$ to derive for all $t \in [0,1]$ that

$$
\begin{aligned}
\mathfrak{l}\big[\mathfrak{g}_{\mathfrak{h}_{\Theta',\Gamma'}[t]}\big] &= \mathfrak{l}\big[\mathfrak{g}_{\mathfrak{h}_{\Theta',\Gamma'}[(1-t)\cdot 0 + t\cdot 1]}\big] \\
&\leq (1-t)\mathfrak{l}\big[\mathfrak{g}_{\mathfrak{h}_{\Theta',\Gamma'}[0]}\big] + t\mathfrak{l}\big[\mathfrak{g}_{\mathfrak{h}_{\Theta',\Gamma'}[1]}\big] \\
&= (1-t)\mathfrak{l}\big[\mathfrak{g}_{\mathfrak{h}_{\Theta',\Gamma'}[0]}\big] + t\mathfrak{l}\big[\mathfrak{g}_{\Gamma'}\big] \\
&\leq (1-t)\mathfrak{l}\big[\mathfrak{g}_{\mathfrak{h}_{\Theta',\Gamma'}[0]}\big] + t\mathfrak{l}\big[\mathfrak{g}_{\mathfrak{h}_{\Theta',\Gamma'}[0]}\big] \\
&= \mathfrak{l}\big[\mathfrak{g}_{\mathfrak{h}_{\Theta',\Gamma'}[0]}\big] .
\end{aligned}
$$

Assume first that the inequality is strict: $\mathfrak{l}[\mathfrak{g}_{\mathfrak{h}_{\Theta',\Gamma'}[\bar{t}]}] < \mathfrak{l}[\mathfrak{g}_{\mathfrak{h}_{\Theta',\Gamma'}[0]}]$ for a $\bar{t} \in [0,1]$. Then, by the assumed convexity of $t \mapsto \mathfrak{l}[\mathfrak{g}_{\mathfrak{h}_{\Theta',\Gamma'}[t]}]$ (see Definition 2) and Lemma 4, there is a $c \in \arg\min_{t \in (0,1]}\{\mathfrak{l}[\mathfrak{g}_{\mathfrak{h}_{\Theta',\Gamma'}[t]}]\}$ such that $\tilde{\mathfrak{h}} : [0,1] \to \mathbb{R}$ defined through $\tilde{\mathfrak{h}}[t] := \mathfrak{l}[\mathfrak{g}_{\mathfrak{h}_{\Theta',\Gamma'}[ct]}]$ for all $t \in [0,1]$ is nonincreasing, $\tilde{\mathfrak{h}}[1]$ is a global minimum of the objective function (4), and $\tilde{\mathfrak{h}}[0] = \mathfrak{l}[\mathfrak{g}_{\mathfrak{h}_{\Theta',\Gamma'}[0]}] = \mathfrak{l}[\mathfrak{g}_{\Theta'}] > \tilde{\mathfrak{h}}[1] = \mathfrak{l}[\mathfrak{g}_{\mathfrak{h}_{\Theta',\Gamma'}[c]}]$. Hence, the function $\bar{\mathfrak{h}} : [0,1] \to \mathcal{M}$ defined through $\bar{\mathfrak{h}}[t] := \mathfrak{h}_{\Theta',\Gamma'}[ct]$ for all $t \in [0,1]$ is a function that satisfies the conditions of Definition 1 for the parameter $\Theta'$ and the global minimum $\tilde{\mathfrak{h}}[1]$. Combining this result with the assumed relationship $\Theta \leftrightarrow \Theta'$ (see Definition 2) yields—see Definition 1—the fact that $\Theta$ is not a spurious local minimum of the objective function (4).

We can thus assume that $t \mapsto \mathfrak{l}[\mathfrak{g}_{\mathfrak{h}_{\Theta',\Gamma'}[t]}]$ is constant, which implies $\Theta' \leftrightarrow \Gamma'$ by Definition 2. The fact that $\Theta \leftrightarrow \Theta'$ (see Definition 2 again) and the transitivity of the path constantness (see Property 3 in Lemma 1) then yield the fact that $\Theta \leftrightarrow \Gamma'$. Hence, $\Theta$ is a global minimum and, therefore, not a spurious local minimum—see Definition 1 again. $\qquad\square$

### A.3 PROOF OF PROPOSITION 2

*Proof of Proposition 2.* Our proof strategy is to apply Lemma 2, which is designed for one individual layer, layer by layer. We first introduce some convenient notation. We define, with some abuse of notation, $\mathfrak{f}^j : \mathbb{R}^{p^j \times n} \to \mathbb{R}^{p^j \times n}$ through $(\mathfrak{f}^j[M])_{uv} := \underline{\mathfrak{f}}^j[M_{uv}]$ for all $j \in \{1,\ldots,l\}$, $u \in \{1,\ldots,p^j\}$, $v \in \{1,\ldots,n\}$, and $M \in \mathbb{R}^{p^j \times n}$. We also define the data matrix $X \in \mathbb{R}^{d \times n}$ through

$X_{ji} := (\boldsymbol{x}_i)_j$ for all $j \in \{1, \dots, d\}$ and $i \in \{1, \dots, n\}$, that is, each column of $X$ consists of one sample. We finally write

$$\mathfrak{g}_{\boldsymbol{\Theta}}[X] := \big(\mathfrak{g}_{\boldsymbol{\Theta}}[\boldsymbol{x}_1], \dots, \mathfrak{g}_{\boldsymbol{\Theta}}[\boldsymbol{x}_n]\big) = \Theta^l \mathfrak{f}^l \big[\Theta^{l-1} \cdots \mathfrak{f}^1[\Theta^0 X]\big] \in \mathbb{R}^{m \times n} \quad \text{for all } \boldsymbol{\Theta} \in \mathcal{M}.$$

Hence, $\mathfrak{g}_{\boldsymbol{\Theta}}[X]$ summarizes the network's outputs for the given data.

Given a parameter $\boldsymbol{\Theta} \in \mathcal{M}$, we establish a corresponding upper-block parameter $\overline{\boldsymbol{\Theta}} \in \mathcal{U}_{s,l}$ layer by layer, starting from the outermost layer. We write

$$\mathfrak{g}_{\boldsymbol{\Theta}}[X] = \underbrace{\Theta^l}_{=:A \in \mathbb{R}^{p^{l+1} \times p^l}} \underbrace{\mathfrak{f}^l}_{=:\mathfrak{h}} \Big[ \underbrace{\Theta^{l-1}}_{=:B \in \mathbb{R}^{p^l \times p^{l-1}}} \underbrace{\mathfrak{f}^{l-1}\big[\Theta^{l-2} \cdots \mathfrak{f}^1[\Theta^0 X]\big]}_{=:C \in \mathbb{R}^{p^{l-1} \times n}} \Big].$$

Lemma 2 for two-layer networks then gives (by Lemma 3, we can assume without loss of generality the fact that $\mathfrak{p}$ is the identity function, that is, $A_{\mathfrak{p}} = A$ and $B^{\mathfrak{p}} = B$)

$$\mathfrak{g}_{\boldsymbol{\Theta}}[X] = \overline{\Theta}^l \mathfrak{f}^l \left[ \begin{pmatrix} \widetilde{\Theta}^{l-1} \\ \mathbf{0} \end{pmatrix} \mathfrak{f}^{l-1}\big[\Theta^{l-2} \cdots \mathfrak{f}^1[\Theta^0 X]\big] \right]$$

for a matrix $\overline{\Theta}^l \in \mathbb{R}^{p^{l+1} \times p^l}$ that satisfies $\underline{\mathfrak{r}}[\overline{\Theta}^l, 1] \leq \underline{\mathfrak{r}}[\Theta^l, 1]$ (recall the definition of $\underline{\mathfrak{r}}$ on Page 7) and meets Condition 3 in the first part of Definition 3 on block parameters as long as $s \geq p^{l+1}(n + 1) = m(n + 1)$, and for a matrix $\widetilde{\Theta}^{l-1} \in \mathbb{R}^{m(n+1) \times p^{l-1}}$ that satisfies $\underline{\mathfrak{r}}[\widetilde{\Theta}^{l-1}, 1] \leq \underline{\mathfrak{r}}[\Theta^{l-1}, 1]$ (or $\underline{\mathfrak{r}}[\widetilde{\Theta}^{l-1}, q] \leq \underline{\mathfrak{r}}[\Theta^{l-1}, q]$ if $l = 1$) and consists of the first $m(n + 1)$ rows of the matrix $\Theta^{l-1}$. (We implicitly assume here and in the following $p^j \geq m(n + 1)^{l-j+1}$ for all $j \in \{1, \dots, l\}$—which is the generic case in view of Corollary 1—to keep the notation manageable, but extending the proof to the general case is straightforward.)

Now, define a parameter $\boldsymbol{\Gamma}^l \in \overline{\mathcal{M}}$ through

$$\boldsymbol{\Gamma}^l := (\overline{\Theta}^l, \Theta^{l-1}, \dots, \Theta^0)$$

and a function $\mathfrak{h}_{\boldsymbol{\Theta}, \boldsymbol{\Gamma}^l} : [0, 1] \to \overline{\mathcal{M}}$ through

$$\mathfrak{h}_{\boldsymbol{\Theta}, \boldsymbol{\Gamma}^l}[t] := (1 - t)\boldsymbol{\Theta} + t\boldsymbol{\Gamma}^l \quad \text{for all } t \in [0, 1].$$

The function $\mathfrak{h}_{\boldsymbol{\Theta}, \boldsymbol{\Gamma}^l}$ is continuous and satisfies $\mathfrak{h}_{\boldsymbol{\Theta}, \boldsymbol{\Gamma}^l}[0] = \boldsymbol{\Theta}$ and $\mathfrak{h}_{\boldsymbol{\Theta}, \boldsymbol{\Gamma}^l}[1] = \boldsymbol{\Gamma}^l$. Moreover, we can 1. use the definitions of the function $\mathfrak{h}_{\boldsymbol{\Theta}, \boldsymbol{\Gamma}^l}$ and the networks, 2. split the network along the outermost layer, 3. invoke the block shape of $\overline{\Theta}^l$ and the definition of $\widetilde{\Theta}^{l-1}$ as the $m(n + 1)$ first rows of the matrix $\Theta^{l-1}$, 4. use the above-stated inequalities for the network $\mathfrak{g}_{\boldsymbol{\Theta}}[X]$, and 5. consolidate the terms to show for all $t \in [0, 1]$ that

$$\begin{aligned}
\mathfrak{g}_{\mathfrak{h}_{\boldsymbol{\Theta}, \boldsymbol{\Gamma}^l}[t]}[X] &= \big((1 - t)\Theta^l + t\overline{\Theta}^l\big)\mathfrak{f}^l\big[\Theta^{l-1} \cdots \mathfrak{f}^1[\Theta^0 X]\big] \\
&= (1 - t)\Theta^l \mathfrak{f}^l\big[\Theta^{l-1} \cdots \mathfrak{f}^1[\Theta^0 X]\big] + t\overline{\Theta}^l \mathfrak{f}^l\big[\Theta^{l-1} \cdots \mathfrak{f}^1[\Theta^0 X]\big] \\
&= (1 - t)\Theta^l \mathfrak{f}^l\big[\Theta^{l-1} \cdots \mathfrak{f}^1[\Theta^0 X]\big] + t\overline{\Theta}^l \mathfrak{f}^l\left[\begin{pmatrix} \widetilde{\Theta}^{l-1} \\ \mathbf{0} \end{pmatrix} \cdots \mathfrak{f}^1[\Theta^0 X]\right] \\
&= (1 - t)\mathfrak{g}_{\boldsymbol{\Theta}}[X] + t\mathfrak{g}_{\boldsymbol{\Theta}}[X] \\
&= \mathfrak{g}_{\boldsymbol{\Theta}}[X].
\end{aligned}$$

Hence, the function $t \mapsto \mathfrak{l}[\mathfrak{g}_{\mathfrak{h}_{\boldsymbol{\Theta}, \boldsymbol{\Gamma}^l}[t]}]$ is constant. Finally, we use 1. the definition of the constraint in (3), 2. the definition of the function $\mathfrak{h}_{\boldsymbol{\Theta}, \boldsymbol{\Gamma}^l}$, 3. the convexity of the $\ell_1$-norm, 4. the above stated fact that $\underline{\mathfrak{r}}[\overline{\Theta}^l, 1] \leq \underline{\mathfrak{r}}[\Theta^l, 1]$, 5. a consolidation, 6. again the definition of the regularizer, and 7. the fact that $\boldsymbol{\Theta} \in \mathcal{M}$ to show for all $t \in [0, 1]$ that

$$\begin{aligned}
\mathfrak{r}\big[\mathfrak{h}_{\boldsymbol{\Theta}, \boldsymbol{\Gamma}^l}[t]\big] &= \max\Big\{a_{\mathfrak{r}} \max_{j \in \{1, \dots, l\}} \big\|\big(\mathfrak{h}_{\boldsymbol{\Theta}, \boldsymbol{\Gamma}^l}[t]\big)^j\big\|_1, b_{\mathfrak{r}}\big\|\big(\mathfrak{h}_{\boldsymbol{\Theta}, \boldsymbol{\Gamma}^l}[t]\big)^0\big\|_q\Big\} \\
&= \max\Big\{a_{\mathfrak{r}} \max_{j \in \{1, \dots, l-1\}} \big\|\Theta^j\big\|_1, a_{\mathfrak{r}}\big\|(1 - t)\Theta^l + t\overline{\Theta}^l\big\|_1, b_{\mathfrak{r}}\big\|\Theta^0\big\|_q\Big\} \\
&\leq \max\Big\{a_{\mathfrak{r}} \max_{j \in \{1, \dots, l-1\}} \big\|\Theta^j\big\|_1, (1 - t)a_{\mathfrak{r}}\big\|\Theta^l\big\|_1 + ta_{\mathfrak{r}}\big\|\overline{\Theta}^l\big\|_1, b_{\mathfrak{r}}\big\|\Theta^0\big\|_q\Big\}
\end{aligned}$$

$$\leq \max\Big\{ a_{\mathfrak{r}} \max_{j\in\{1,\ldots,l-1\}} \|\Theta^j\|_1, (1-t)a_{\mathfrak{r}}\|\Theta^l\|_1 + ta_{\mathfrak{r}}\|\Theta^l\|_1, b_{\mathfrak{r}}\|\Theta^0\|_q \Big\}$$

$$= \max\Big\{ a_{\mathfrak{r}} \max_{j\in\{1,\ldots,l-1\}} \|\Theta^j\|_1, a_{\mathfrak{r}}\|\Theta^l\|_1, b_{\mathfrak{r}}\|\Theta^0\|_q \Big\}$$

$$= \mathfrak{r}[\Theta]$$

$$\leq 1 .$$

Hence, $\mathfrak{h}_{\Theta,\Gamma^l}[t] \in \mathcal{M}$ for all $t \in [0,1]$. In conclusion, we have shown—see Definition 2—that $\Theta$ and $\Gamma^l$ are path constant: $\Theta \leftrightarrow \Gamma^l$.

We then move one layer inward. Lemma 2 ensures that (recall again Lemma 3)

$$\underbrace{\widetilde{\Theta}^{l-1}}_{A\in\mathbb{R}^{m(n+1)\times p^{l-1}}} \underbrace{\mathfrak{f}^{l-1}}_{\mathfrak{h}}[\underbrace{\Theta^{l-2}}_{B\in\mathbb{R}^{p^{l-1}\times p^{l-2}}} \underbrace{\cdots \mathfrak{f}^1[\Theta^0 X]]}_{C\in\mathbb{R}^{p^{l-2}\times n}} = \dot{\Theta}^{l-1}\mathfrak{f}^{l-1}\left[ \begin{pmatrix} \widetilde{\Theta}^{l-2} \\ \mathbf{0} \end{pmatrix} \cdots \mathfrak{f}^1[\Theta^0 X] \right]$$

for a matrix $\dot{\Theta}^{l-1} \in \mathbb{R}^{m(n+1)\times p^{l-1}}$ that satisfies $\underline{\mathfrak{r}}[\dot{\Theta}^{l-1},1] \leq \underline{\mathfrak{r}}[\widetilde{\Theta}^{l-1},1] \leq \underline{\mathfrak{r}}[\Theta^{l-1},1]$ and meets Condition 3 in the first part of Definition 3 on block parameters as long as $s \geq m(n+1)^2$, and for a matrix $\widetilde{\Theta}^{l-2} \in \mathbb{R}^{m(n+1)^2 \times p^{l-2}}$ that satisfies $\underline{\mathfrak{r}}[\widetilde{\Theta}^{l-2},1] \leq \underline{\mathfrak{r}}[\Theta^{l-2},1]$ (or $\underline{\mathfrak{r}}[\widetilde{\Theta}^{l-2},q] \leq \underline{\mathfrak{r}}[\Theta^{l-2},q]$ if $l=2$) and consists of the first $m(n+1)^2$ rows of the matrix $\Theta^{l-2}$.

Next, we define $\overline{\Theta}^{l-1} \in \mathbb{R}^{p^l \times p^{l-1}}$ through $(\overline{\Theta}^{l-1})_{uv} := (\dot{\Theta}^{l-1})_{uv}$ for $u \leq m(n+1)$ and $(\overline{\Theta}^{l-1})_{uv} := 0$ otherwise. Combining this definition with the above-derived results yields

$$\mathfrak{g}_\Theta[X] = \overline{\Theta}^l \mathfrak{f}^l\left[ \overline{\Theta}^{l-1}\mathfrak{f}^{l-1}\left[ \begin{pmatrix} \widetilde{\Theta}^{l-2} \\ \mathbf{0} \end{pmatrix} \cdots \mathfrak{f}^1[\Theta^0 X] \right] \right] ,$$

and the matrix $\overline{\Theta}^{l-1}$ satisfies $\underline{\mathfrak{r}}[\overline{\Theta}^{l-1},1] = \underline{\mathfrak{r}}[\dot{\Theta}^{l-1},1] \leq \underline{\mathfrak{r}}[\Theta^{l-1},1]$ and meets Condition 2 in the first part of Definition 3 on block parameters as long as $s \geq m(n+1)^2$.

Similarly as above, define a parameter $\Gamma^{l-1} \in \overline{\mathcal{M}}$ through

$$\Gamma^{l-1} := (\overline{\Theta}^l, \overline{\Theta}^{l-1}, \Theta^{l-2}, \ldots, \Theta^0)$$

and a function $\mathfrak{h}_{\Gamma^l,\Gamma^{l-1}} : [0,1] \to \overline{\mathcal{M}}$ through

$$\mathfrak{h}_{\Gamma^l,\Gamma^{l-1}}[t] := (1-t)\Gamma^l + t\Gamma^{l-1} \qquad \text{for all } t \in [0,1]$$

to show that $\Gamma^l \leftrightarrow \Gamma^{l-1}$. In view of Property 3 in Lemma 1, we can conclude that $\Theta \leftrightarrow \Gamma^{l-1}$.

Finish the proof by induction over the layers, and note that the lower-block parameters can be established in the same way. □

## A.4 PROOF OF PROPOSITION 3

*Proof of Proposition 3.* The key ingredient of the proof is the assumed block structure of the parameters. Consider two block parameters $\Theta \in \mathcal{U}_{s,l}$ and $\Gamma \in \mathcal{L}_{s,l}$ and define the parameters

$$\Theta' := (\Theta^l, \Theta^{l-1} + \Gamma^{l-1}, \ldots, \Theta^0 + \Gamma^0) \in \overline{\mathcal{M}};$$

$$\Gamma' := (\Gamma^l, \Theta^{l-1} + \Gamma^{l-1}, \ldots, \Theta^0 + \Gamma^0) \in \overline{\mathcal{M}}$$

and the function

$$\mathfrak{h}_{\Theta',\Gamma'} : [0,1] \to \overline{\mathcal{M}}$$

$$t \mapsto (1-t)\Theta' + t\Gamma' = \big((1-t)\Theta^l + t\Gamma^l, \Theta^{l-1} + \Gamma^{l-1}, \ldots, \Theta^0 + \Gamma^0\big) .$$

By the row-wise structure of the constraint (see Page 3 again), the convexity of the $\ell_1$-norm, and the block shapes of the parameters (see Figure 2 again), we can find that $\mathfrak{r}[\mathfrak{h}_{\Theta',\Gamma'}[t]] \leq \max\{\mathfrak{r}[\Theta], \mathfrak{r}[\Gamma]\} \leq 1$ for all $t \in [0,1]$, that is, $\mathfrak{h}_{\Theta',\Gamma'}[t] \in \mathcal{M}$ for all $t \in [0,1]$. One can also verify readily the fact that the function $\mathfrak{h}_{\Theta',\Gamma'}$ is continuous, $\mathfrak{h}_{\Theta',\Gamma'}[0] = \Theta'$, and $\mathfrak{h}_{\Theta',\Gamma'}[1] = \Gamma'$. Next, we define

$$[\Theta', \Gamma']_{c_1,c_2} := (c_1\Theta^l + c_2\Gamma^l, \Theta^{l-1} + \Gamma^{l-1}, \ldots, \Theta^0 + \Gamma^0) \in \overline{\mathcal{M}} \qquad \text{for all } c_1, c_2 \in \mathbb{R},$$

which generalizes $\mathfrak{h}_{\Theta',\Gamma'}$ in the sense that $\mathfrak{h}_{\Theta',\Gamma'}[t] = [\Theta',\Gamma']_{1-t,t}$ for all $t \in [0,1]$. We then 1. invoke the definition of $[\Theta',\Gamma']_{c_1,c_2}$ and the definition of the networks in (1), 2. split the network along the outer layer, 3. use the block structures of the parameters and the assumption that $\mathfrak{f}^j[b] = (\underline{\mathfrak{f}}^j[b_1],\ldots,\underline{\mathfrak{f}}^j[b_{p^j}])^\top$ for all $j \in \{1,\ldots,l\}$ and $b \in \mathbb{R}^{p^j}$, 4. continue in this fashion, and 5. invoke again the definition of the networks in (1) to find for all $c_1, c_2 \in \mathbb{R}$ and $x \in \mathcal{D}_x$ that

$$
\mathfrak{g}_{[\Theta',\Gamma']_{c_1,c_2}}[x]
$$
$$
= (c_1\Theta^l + c_2\Gamma^l)\mathfrak{f}^l\left[(\Theta^{l-1} + \Gamma^{l-1})\mathfrak{f}^{l-1}\left[\cdots\mathfrak{f}^1\left[(\Theta^0 + \Gamma^0)x\right]\right]\right]
$$
$$
= c_1\Theta^l\mathfrak{f}^l\left[(\Theta^{l-1} + \Gamma^{l-1})\mathfrak{f}^{l-1}\left[\cdots\mathfrak{f}^1\left[(\Theta^0 + \Gamma^0)x\right]\right]\right]
$$
$$
+ c_2\Gamma^l\mathfrak{f}^l\left[(\Theta^{l-1} + \Gamma^{l-1})\mathfrak{f}^{l-1}\left[\cdots\mathfrak{f}^1\left[(\Theta^0 + \Gamma^0)x\right]\right]\right]
$$
$$
= c_1\Theta^l\mathfrak{f}^l\left[\Theta^{l-1}\mathfrak{f}^{l-1}\left[\cdots\mathfrak{f}^1\left[(\Theta^0 + \Gamma^0)x\right]\right]\right] + c_2\Gamma^l\mathfrak{f}^l\left[\Gamma^{l-1}\mathfrak{f}^{l-1}\left[\cdots\mathfrak{f}^1\left[(\Theta^0 + \Gamma^0)x\right]\right]\right]
$$
$$
= \cdots = c_1\Theta^l\mathfrak{f}^l\left[\Theta^{l-1}\mathfrak{f}^{l-1}\left[\cdots\mathfrak{f}^1\left[\Theta^0 x\right]\right]\right] + c_2\Gamma^l\mathfrak{f}^l\left[\Gamma^{l-1}\mathfrak{f}^{l-1}\left[\cdots\mathfrak{f}^1\left[\Gamma^0 x\right]\right]\right]
$$
$$
= c_1\mathfrak{g}_\Theta[x] + c_2\mathfrak{g}_\Gamma[x].
$$

Finally, we use 1. the definition of the function $\mathfrak{h}_{\Theta',\Gamma'}$ and of the parameter $[\Theta',\Gamma']_{c_1,c_2}$, 2. the above display with $c_1 = 1 - (1-a)t_1 - at_2$ and $c_2 = (1-a)t_1 + at_2$, 3. a rearrangement of the terms, 4. the assumed convexity of the loss function $\mathfrak{l}$, 5. again the above display, and 6. again the definition of $\mathfrak{h}_{\Theta',\Gamma'}$, to find for all $a, t_1, t_2 \in [0,1]$ that

$$
\mathfrak{l}\left[\mathfrak{g}_{\mathfrak{h}_{\Theta',\Gamma'}[(1-a)t_1+at_2]}\right]
$$
$$
= \mathfrak{l}\left[\mathfrak{g}_{[\Theta',\Gamma']_{1-(1-a)t_1-at_2,(1-a)t_1+at_2}}\right]
$$
$$
= \mathfrak{l}\left[\left(1 - (1-a)t_1 - at_2\right)\mathfrak{g}_\Theta + \left((1-a)t_1 + at_2\right)\mathfrak{g}_\Gamma\right]
$$
$$
= \mathfrak{l}\left[\left((1-a) - (1-a)t_1\right)\mathfrak{g}_\Theta + (a - at_2)\mathfrak{g}_\Theta + (1-a)t_1\mathfrak{g}_\Gamma + at_2\mathfrak{g}_\Gamma\right]
$$
$$
= \mathfrak{l}\left[(1-a)\left((1-t_1)\mathfrak{g}_\Theta + t_1\mathfrak{g}_\Gamma\right) + a\left((1-t_2)\mathfrak{g}_\Theta + t_2\mathfrak{g}_\Gamma\right)\right]
$$
$$
\leq (1-a)\mathfrak{l}\left[(1-t_1)\mathfrak{g}_\Theta + t_1\mathfrak{g}_\Gamma\right] + a\mathfrak{l}\left[(1-t_2)\mathfrak{g}_\Theta + t_2\mathfrak{g}_\Gamma\right]
$$
$$
= (1-a)\mathfrak{l}\left[\mathfrak{g}_{[\Theta',\Gamma']_{1-t_1,t_1}}\right] + a\mathfrak{l}\left[\mathfrak{g}_{[\Theta',\Gamma']_{1-t_2,t_2}}\right]
$$
$$
= (1-a)\mathfrak{l}\left[\mathfrak{g}_{\mathfrak{h}_{\Theta',\Gamma'}[t_1]}\right] + a\mathfrak{l}\left[\mathfrak{g}_{\mathfrak{h}_{\Theta',\Gamma'}[t_2]}\right],
$$

which means that $t \mapsto \mathfrak{l}[\mathfrak{g}_{\mathfrak{h}_{\Theta',\Gamma'}[t]}]$ is convex. We conclude—see Definition 2—that $\Theta' \smile \Gamma'$.

In view of Definition 2, it is left to show that $\Theta \leftrightarrow \Theta'$ and $\Gamma \leftrightarrow \Gamma'$. Consider now the function

$$
\mathfrak{h}_{\Theta,\Theta'} : [0,1] \to \overline{\mathcal{M}}
$$
$$
t \mapsto (\Theta^l, \Theta^{l-1} + t\Gamma^{l-1}, \ldots, \Theta^0 + t\Gamma^0).
$$

By the row-wise structure of the constraint (see Page 3 once more) and the block shapes of the parameters (see Figure 2 once more), we can find that $\mathfrak{r}[\mathfrak{h}_{\Theta,\Theta'}[t]] \leq \max\{\mathfrak{r}[\Theta], \mathfrak{r}[t\Gamma]\} \leq 1$ for all $t \in [0,1]$, that is, $\mathfrak{h}_{\Theta,\Theta'}[t] \in \mathcal{M}$ for all $t \in [0,1]$. One can also verify readily the fact that the function $\mathfrak{h}_{\Theta,\Theta'}$ is continuous, $\mathfrak{h}_{\Theta,\Theta'}[0] = \Theta$, and $\mathfrak{h}_{\Theta,\Theta'}[1] = \Theta'$. Moreover, we can 1. invoke the definition of $\mathfrak{h}_{\Theta,\Theta'}$ and the definition of the networks in (1), 2. use the block structures of the parameters and the elementwise structure of the activation function, 3. continue in this fashion, and 4. invoke the definitions of the function $\mathfrak{h}_{\Theta,\Theta'}$ and the networks in (1) again to find for all $t \in [0,1]$ and $x \in \mathcal{D}_x$ that

$$
\mathfrak{g}_{\mathfrak{h}_{\Theta,\Theta'}[t]}[x] = \Theta^l\mathfrak{f}^l\left[(\Theta^{l-1} + t\Gamma^{l-1})\mathfrak{f}^{l-1}\left[\cdots\mathfrak{f}^1\left[(\Theta^0 + t\Gamma^0)x\right]\right]\right]
$$
$$
= \Theta^l\mathfrak{f}^l\left[\Theta^{l-1}\mathfrak{f}^{l-1}\left[\cdots\mathfrak{f}^1\left[(\Theta^0 + t\Gamma^0)x\right]\right]\right]
$$

$$= \cdots = \Theta^l \mathfrak{f}^l \left[ \Theta^{l-1} \mathfrak{f}^{l-1} \left[ \cdots \mathfrak{f}^1 \left[ \Theta^0 \boldsymbol{x} \right] \right] \right]$$

$$= \mathfrak{g}_{\mathfrak{h}_{\Theta, \Theta'}[0]}[\boldsymbol{x}] \,,$$

which implies that the function $t \mapsto \mathfrak{l}[\mathfrak{g}_{\mathfrak{h}_{\Theta, \Theta'}[t]}]$ is constant. We conclude that—see Definition 2—that $\Theta \leftrightarrow \Theta'$.

We can show in a similar way that $\boldsymbol{\Gamma} \leftrightarrow \boldsymbol{\Gamma}'$. Hence, given Definition 2, we find that $\Theta \leftrightsquigarrow \boldsymbol{\Gamma}$, as desired. $\qquad \square$

### A.5 Proof of Lemma 2

*Proof of Lemma 2.* The proof is essentially a careful reparametrization. We proceed in three steps: see Figure 4 for an overview.

*Step 1:* Fix a $k \in \{1, \ldots, u\}$. We first show that there is a matrix $\dot{A} \in \mathbb{R}^{u \times v}$ such that

1. $\dot{A}\mathfrak{h}[BC] = A\mathfrak{h}[BC]$;

2. $\mathfrak{r}[\dot{A}, q_A] \leq \mathfrak{r}[A, q_A]$;

3. $\#\{j \in \{1, \ldots, v\} : \dot{A}_{kj} \neq 0\} \leq r + 1$;

4. $\dot{A}_{aj} = A_{aj}$ for all $a \neq k$.

Hence, we replace the matrix $A$, which contains the "output parameters," by a matrix whose $k$th row has at most $r + 1$ nonzero entries—see the illustration in Figure 4.

The proof of this step is based on our version of Carathéodory's theorem in Lemma 5. For every $k \in \{1, \ldots, u\}$ and $i \in \{1, \ldots, r\}$, elementary matrix algebra yields that

$$\left(A\mathfrak{h}[BC]\right)_{ki} = \sum_{j=1}^{v} A_{kj} \left(\mathfrak{h}[BC]\right)_{ji} \,.$$

Denoting the row vectors of a matrix $M \in \mathbb{R}^{a \times b}$ by $M_{1\bullet}, \ldots, M_{a\bullet} \in \mathbb{R}^b$, we then get

$$\left(A\mathfrak{h}[BC]\right)_{k\bullet} = \sum_{j=1}^{v} A_{kj} \left(\mathfrak{h}[BC]\right)_{j\bullet} \,.$$

The case $\|A_{k\bullet}\|_{q_A} = 0$ is straightforward to deal with: all elements of $A_{k\bullet}$ are then equal to zero, and we can just set $\dot{A} := A$.

We can thus assume the fact that $\|A_{k\bullet}\|_{q_A} \neq 0$. We then get from the preceding equality that

$$\left(A\mathfrak{h}[BC]\right)_{k\bullet} = \sum_{j=1}^{v} \underbrace{\frac{A_{kj}}{\|A_{k\bullet}\|_{q_A}}}_{=: \tilde{t}_j} \underbrace{\|A_{k\bullet}\|_{q_A} \left(\mathfrak{h}[BC]\right)_{j\bullet}}_{=: \boldsymbol{z}_j} \,.$$

Since

$$\|\tilde{\boldsymbol{t}}\|_{q_A} = \left(\sum_{j=1}^{v} |\tilde{t}_j|^{q_A}\right)^{1/q_A} = \left(\sum_{j=1}^{v} \left|\frac{A_{kj}}{\|A_{k\bullet}\|_{q_A}}\right|^{q_A}\right)^{1/q_A}$$

$$= \frac{1}{\|A_{k\bullet}\|_{q_A}} \left(\sum_{j=1}^{v} |A_{kj}|^{q_A}\right)^{1/q_A} = \frac{\|A_{k\bullet}\|_{q_A}}{\|A_{k\bullet}\|_{q_A}} = 1 \,,$$

the previous equality means that the vector $\left(A\mathfrak{h}[BC]\right)_{k\bullet} \in \mathbb{R}^r$ is a $q_A$-convex combination of the $2v$ vectors $\boldsymbol{z}_1, \ldots, \boldsymbol{z}_v, -\boldsymbol{z}_1, \ldots, -\boldsymbol{z}_v \in \mathbb{R}^r$, that is, $\left(A\mathfrak{h}[BC]\right)_{k\bullet} \in \mathrm{conv}_q[\boldsymbol{z}_1, \ldots, \boldsymbol{z}_v, -\boldsymbol{z}_1, \ldots, -\boldsymbol{z}_v]$.

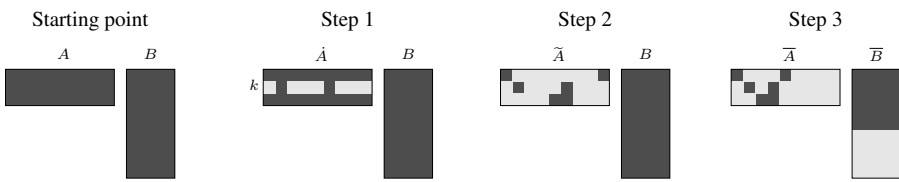

Figure 4: overview of the proof of Lemma 2

Hence, by Lemma 5, there is a $t \in [-1,1]^v$ such that $\|t\|_{q_A} \leq 1$, $\#\{j \in \{1,\ldots,v\} : t_j \neq 0\} \leq r+1$, and

$$\left(A\mathfrak{h}[BC]\right)_{k\bullet} = \sum_{j=1}^{v} t_j \|A_{k\bullet}\|_{q_A} \left(\mathfrak{h}[BC]\right)_{j\bullet}.$$

Hence, by the definition of the row vectors,

$$\left(A\mathfrak{h}[BC]\right)_{ki} = \sum_{j=1}^{v} t_j \|A_{k\bullet}\|_{q_A} \left(\mathfrak{h}[BC]\right)_{ji}$$

for all $i \in \{1,\ldots,r\}$, and, more generally, we find for all $a \in \{1,\ldots,u\}$ and $i \in \{1,\ldots,r\}$ that

$$\left(A\mathfrak{h}[BC]\right)_{ai} = \begin{cases} \sum_{j=1}^{v} t_j \|A_{k\bullet}\|_{q_A} \left(\mathfrak{h}[BC]\right)_{ji} & \text{for } a = k\,; \\ \sum_{j=1}^{v} A_{aj} \left(\mathfrak{h}[BC]\right)_{ji} & \text{otherwise}. \end{cases}$$

This motivates us to define $\dot{A} \in \mathbb{R}^{u \times v}$ through

$$\dot{A}_{aj} := \begin{cases} t_j \|A_{k\bullet}\|_{q_A} & \text{for } a = k\,; \\ A_{aj} & \text{otherwise}. \end{cases}$$

Properties 1, 3, and 4 then follow immediately. Property 2 can be derived by using 1. the definition of the basic regularizer $\underline{\mathfrak{r}}$ on Page 7, 2. the definition of $\dot{A}$, 3. the linearity of finite sums, 4. the definition of the $\ell_q$-norms on Page 8, 5. the above-derived property $\|t\|_{q_A} \leq 1$, 6. a consolidation, and 7. again the definition of the basic regularizer $\underline{\mathfrak{r}}$ on Page 7:

$$\left(\underline{\mathfrak{r}}[\dot{A}, q_A]\right)^{q_A} = \max_{a \in \{1,\ldots,u\}} \sum_{j=1}^{v} |\dot{A}_{aj}|^{q_A}$$

$$= \max_{a \in \{1,\ldots,u\}} \begin{cases} \sum_{j=1}^{v} \left|t_j \|A_{k\bullet}\|_{q_A}\right|^{q_A} & \text{for } a = k \\ \sum_{j=1}^{v} |A_{aj}|^{q_A} & \text{otherwise} \end{cases}$$

$$= \max_{a \in \{1,\ldots,u\}} \begin{cases} \|A_{k\bullet}\|_{q_A}^{q_A} \sum_{j=1}^{v} |t_j|^{q_A} & \text{for } a = k \\ \sum_{j=1}^{v} |A_{aj}|^{q_A} & \text{otherwise} \end{cases}$$

$$= \max_{a \in \{1,\ldots,u\}} \begin{cases} \left(\sum_{j=1}^{v} |A_{kj}|^{q_A}\right) \|t\|_{q_A}^{q_A} & \text{for } a = k \\ \sum_{j=1}^{v} |A_{aj}|^{q_A} & \text{otherwise} \end{cases}$$

$$\leq \max_{a \in \{1,\ldots,u\}} \begin{cases} \sum_{j=1}^{v} |A_{kj}|^{q_A} & \text{for } a = k \\ \sum_{j=1}^{v} |A_{aj}|^{q_A} & \text{otherwise} \end{cases}$$

$$= \max_{a \in \{1,\ldots,u\}} \sum_{j=1}^{v} |A_{aj}|^{q_A}$$

$$= \left(\underline{\mathfrak{r}}[A, q_A]\right)^{q_A},$$

as desired. This concludes the proof of the first step.

*Step 2:* We now show that there is a matrix $\widetilde{A} \in \mathbb{R}^{u \times v}$ such that

1. $\widetilde{A}\mathfrak{h}[BC] = A\mathfrak{h}[BC]$;

2. $\mathfrak{r}[\widetilde{A}, q_A] \leq \mathfrak{r}[A, q_A]$;

3. $\#\{j \in \{1, \ldots, v\} : \widetilde{A}_{aj} \neq 0\} \leq r + 1$ for all $a \in \{1, \ldots, u\}$.

Hence, we replace the matrix $A$ by a matrix whose every row has at most $r + 1$ nonzero entries—see again the illustration in Figure 4.

Since Step 1 changes only the $k$th row of $A$ (see Property 4 derived in Step 1), we can apply it to one row after another.

*Step 3:* We finally prove the first part of the lemma—see again the illustration in Figure 4.

By Property 3 of the previous step, the matrix $\widetilde{A}$ has at most $u(r + 1)$ nonzero columns. Verify that replacing $A$ by $A_{\mathfrak{p}}$ and $B$ by $B^{\mathfrak{p}}$ for a suitable permutation $\mathfrak{p}$ leads to an $\widetilde{A}$ whose entries outside the first $u(r + 1)$ columns are equal to zero—while all other properties remain intact. We denote this version of $\widetilde{A}$ by $\overline{A}$. We then derive for all $j \in \{1, \ldots, v\}$ and $i \in \{1, \ldots, r\}$ that

$$\left(\mathfrak{h}[B^{\mathfrak{p}}C]\right)_{ji} = \mathfrak{h}\left[(B^{\mathfrak{p}}C)_{ji}\right] = \mathfrak{h}\left[\sum_{b=1}^{o}(B^{\mathfrak{p}})_{jb}C_{bi}\right] = \mathfrak{h}\left[\sum_{b=1}^{o}(C^{\top})_{ib}(B^{\mathfrak{p}})_{jb}\right]$$
$$= \mathfrak{h}\left[(C^{\top}(B^{\mathfrak{p}})_{j\bullet})_i\right] = \left(\mathfrak{h}[C^{\top}(B^{\mathfrak{p}})_{j\bullet}]\right)_i,$$

where we define (with some abuse of notation) $\mathfrak{h} : \mathbb{R}^r \to \mathbb{R}^r$ through $(\mathfrak{h}[\boldsymbol{b}])_i := \mathfrak{h}[b_i]$ for all $\boldsymbol{b} \in \mathbb{R}^r$. Combining this result with the results of Step 2 (with $A$ and $B$ replaced by $A_{\mathfrak{p}}$ and $B^{\mathfrak{p}}$, respectively) yields for all $a \in \{1, \ldots, u\}$ and $i \in \{1, \ldots, r\}$ that

$$\left(A_{\mathfrak{p}}\mathfrak{h}[B^{\mathfrak{p}}C]\right)_{ai} = \sum_{j=1}^{v}(A_{\mathfrak{p}})_{aj}\left(\mathfrak{h}[B^{\mathfrak{p}}C]\right)_{ji} = \sum_{j=1}^{v}(A_{\mathfrak{p}})_{aj}\left(\mathfrak{h}[C^{\top}(B^{\mathfrak{p}})_{j\bullet}]\right)_i$$
$$= \sum_{j=1}^{v}\overline{A}_{aj}\left(\mathfrak{h}[C^{\top}(B^{\mathfrak{p}})_{j\bullet}]\right)_i = \sum_{j=1}^{\min\{u(r+1),v\}}\overline{A}_{aj}\left(\mathfrak{h}[C^{\top}(B^{\mathfrak{p}})_{j\bullet}]\right)_i.$$

We then define $\overline{B} \in \mathbb{R}^{v \times o}$ through

$$\overline{B}_{ji} := \begin{cases} (B^{\mathfrak{p}})_{ji} = B_{\mathfrak{p}[j]i} & \text{for } j \leq u(r + 1); \\ 0 & \text{otherwise}. \end{cases}$$

We then use 1. the above-stated equality, 2. the definition of $\overline{B}$, 3. a similar derivation as above, 4. the block structure of $\overline{A}$, and 5. a similar derivation as above to establish for all $a \in \{1, \ldots, u\}$ and $i \in \{1, \ldots, r\}$ the fact that

$$\left(A_{\mathfrak{p}}\mathfrak{h}[B^{\mathfrak{p}}C]\right)_{ai} = \sum_{j=1}^{\min\{u(r+1),v\}}\overline{A}_{aj}\left(\mathfrak{h}[C^{\top}(B^{\mathfrak{p}})_{j\bullet}]\right)_i$$
$$= \sum_{j=1}^{\min\{u(r+1),v\}}\overline{A}_{aj}\left(\mathfrak{h}[C^{\top}\overline{B}_{j\bullet}]\right)_i$$
$$= \sum_{j=1}^{\min\{u(r+1),v\}}\overline{A}_{aj}\left(\mathfrak{h}[\overline{B}C]\right)_{ji}$$
$$= \sum_{j=1}^{v}\overline{A}_{aj}\left(\mathfrak{h}[\overline{B}C]\right)_{ji}$$
$$= \left(\overline{A}\mathfrak{h}[\overline{B}C]\right)_{ai}.$$

The other properties stated in the lemma follow readily.

The second part of the lemma can be derived in the same way. □

## A.6   PROOF OF LEMMA 4

*Proof of Lemma 4.* The proof is a simple exercise in calculus. An illustration of the quantities involved in the proof is given in Figure 5. The function $\mathfrak{h}$ is convex by assumption; hence, it is continuous. Then, according to the extreme value theorem, there is a number

$$c \in \underset{t \in [0,1]}{\arg\min} \big\{ \mathfrak{h}[t] \big\}.$$

Since $\mathfrak{h}[\bar{t}] < \mathfrak{h}[0]$ for a $\bar{t} \in (0,1]$, it holds that $c \in (0,1]$. Now consider $t_2 \in [0,1]$ and $t_1 \in [0,t_2)$, that is, $t_2 > t_1$. Basic calculus ensures that

$$
\begin{aligned}
\tilde{\mathfrak{h}}[t_2] &= \mathfrak{h}[ct_2] \\
&= \mathfrak{h}\big[ ct_1 + ct_2 - ct_1 \big] \\
&= \mathfrak{h}\left[ ct_1 + \frac{ct_2 - ct_1}{c - ct_1}(c - ct_1) \right] \\
&= \mathfrak{h}\left[ ct_1 + \frac{t_2 - t_1}{1 - t_1}(c - ct_1) \right] \\
&= \mathfrak{h}\left[ \left(1 - \frac{t_2 - t_1}{1 - t_1}\right)ct_1 + \frac{t_2 - t_1}{1 - t_1}c \right] \\
&\leq \left(1 - \frac{t_2 - t_1}{1 - t_1}\right)\mathfrak{h}[ct_1] + \frac{t_2 - t_1}{1 - t_1}\mathfrak{h}[c] \\
&\leq \left(1 - \frac{t_2 - t_1}{1 - t_1}\right)\mathfrak{h}[ct_1] + \frac{t_2 - t_1}{1 - t_1}\mathfrak{h}[ct_1] \\
&= \mathfrak{h}[ct_1] \\
&= \tilde{\mathfrak{h}}[t_1].
\end{aligned}
$$

Hence, $\tilde{\mathfrak{h}}$ is nonincreasing.

Moreover,

$$\tilde{\mathfrak{h}}[0] = \mathfrak{h}[c \cdot 0] = \mathfrak{h}[0] > \mathfrak{h}[\bar{t}] \geq \mathfrak{h}[c] = \mathfrak{h}[c \cdot 1] = \tilde{\mathfrak{h}}[1].$$

Hence, $\tilde{\mathfrak{h}}[0] > \tilde{\mathfrak{h}}[1]$. This concludes the proof.  □

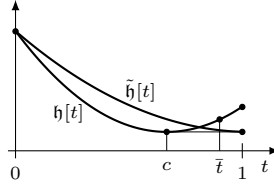

Figure 5: quantities in the proof of Lemma 4

