# OpenReview forum: "No Spurious Local Minima: on the Optimization Landscapes of Wide and Deep Neural Networks"
_ICLR.cc/2021/Conference — Reject_

### Official Review · AnonReviewer3 · 2020-10-28
**An interesting but not surprising result**

**Rating:** 4
**Confidence:** 4

**Review:**

It is still a myth about why neural networks can work without convexity (or, with many local minima).
Some existing papers tried to explain it using “spurious” minima, which means that there is no non-increasing path to a global minimum.
This paper proved that the deep neural networks have no spurious minima with large width (>=2m(n+1)^l, l=#depth, n=#samples, m=#output dim), which is an extension of Venturi (2019) and Lacotte & Pilanci (2020).
Compared with the previous works of two-layer neural networks, this paper considered deeper neural networks.
However, I had several concerns about the conclusion and the techniques. Please refer to CONCERNS for more details.

### CONTRIBUTIONS
a) This paper considers a general setting, including several activation functions (including ReLU) and constraint and unconstraint estimation.

b) This paper considers deeper neural networks instead of two-layer neural networks, which is an extension of the existing works.

### CONCERNS
To be honest, I did not check all the details in the proof. But I have several concerns:

1. The major concern comes from the conclusion. I notice that the depth $l$ appears in the power. It seems that the bound is not tight and surprising enough.

2. If the bound is tight, one needs wider neural networks in deeper neural networks. I do not think that is the reality.

3. As for the technical part, I am afraid I missed some details. But after checking the proof, I found that the proof is finished by induction (page 13). It seems a little bit incremental. By the way, that is also why $l$ appears in the power.

4. Therefore, I would like to raise my score if the authors can restate their technical novelty, or at least show more about the importance of the conclusions.

### COMMENTS
Overall, this is an interesting paper. But I am still concerned with its technical novelty and the conclusion importance. I tend to reject this paper without a convincing response.

---

### Official Review · AnonReviewer2 · 2020-10-29

**Rating:** 5
**Confidence:** 4

**Review:**

Summary

This paper studies the optimization landscape of the training loss of deep neural networks. For  a general setup, the paper shows that if the network width is greater than $2m(n+1)^l$, then any parameter value has a path to a global minimum on which the training loss does not increase. Here, $m$ is the output dimension, $n$ is the number of training examples, and $l$ is the number of hidden layers.

Strength

The result presented in the paper holds for a surprisingly wide range of setups. The theorem holds for any convex loss function, any arbitrary activation, arbitrary output dimension and depth. The theorem holds for unconstrained optimization setup, and also some specific type of constrained setup (Eq (3)).

The paper is well-written and it delivers its key messages fairly well. The main text provides a good proof sketch that is easy to follow. After a quick perusal of the proof I am convinced that the proof is correct. Overall, I enjoyed reading this paper.

Weakness

There are two main weaknesses of this paper: one is the huge width requirement that exponentially grows with depth, the other is too many missing (directly relevant) citations and the unconventional usage of the term “spurious local minima”.

Let me start with the second one. There is a large body of literature that refers to non-optimal local minima as spurious/bad local minima, and investigates existence/nonexistence of such local minima in the context of neural networks [1, 2, 3, 4, and many more]. In fact, for deep networks with piecewise linear activations, no matter how wide the network is, it is known that there are non-optimal local minima in the training loss in the unconstrained case [3, 4]. The paper misses the entire body of literature and defines spurious local minima in a different way, which can confuse the readers. The title already sounds like a contradiction to [3, 4]!

More importantly, there are also other missing citations that are even more directly relevant, namely, the ones studying nonincreasing paths to global minima [5, 6, 7]. This set of results typically require that one of the hidden layers is wider than the number of data points $n$ and that the subsequent layers after the wide hidden layer only get narrower. Although these results hold for more specific setups than this paper, the difference in the width requirement is huge: $n$ vs $O(n^l)$.

In light of these existing results, I fear that the width requirement in this paper is too big, and it grows exponentially with the depth $l$. Although I liked the construction illustrated in this paper, this significant weakness of the main result makes me hesitate to recommend acceptance.

One minor weakness I also wanted to point out was that, “nonincreasing path to global minimum” alone is not enough, in the sense that this property does not rule out the existence of a local minimum in a locally constant region of the loss landscape (think of the case where all the ReLUs are turned off in a ReLU network). Although sufficient width can ensure the existence of a nonincreasing path to a global minimum from this locally constant region, there is no way that a gradient-based local search algorithm can escape this region.

Overall Assessment

Although I liked this paper as I read it, I believe the weaknesses of the main result are too significant. I also think that the paper could use a rewriting to contextualize the results relative to the missing citations. I lean slightly towards rejection at this time.


[1] Kenji Kawaguchi. Deep Learning without Poor Local Minima, 2016

[2] Itay Safran, Ohad Shamir. Spurious Local Minima are Common in Two-Layer ReLU Neural Networks, 2018

[3] Chulhee Yun, Suvrit Sra, Ali Jadbabaie. Small nonlinearities in activation functions create bad local minima in neural networks, 2019

[4] Fengxiang He, Bohan Wang, Dacheng Tao. Piecewise linear activations substantially shape the loss surfaces of neural networks, 2020

[5] Benjamin D. Haeffele, Rene Vidal. Global optimality in neural network training, 2017

[6] Quynh Nguyen. On Connected Sublevel Sets in Deep Learning, 2019

[7] Henning Petzka, Cristian Sminchisescu. Non-attracting Regions of Local Minima in Deep and Wide Neural Networks, 2018

---

### Official Review · AnonReviewer1 · 2020-11-01
**An interesting work, but the settings/results are far from practice**

**Rating:** 4
**Confidence:** 3

**Review:**

This work showed that for a wide deep neural network, the optimization landscapes of empirical-risk minimizers over wide feedforward networks have no spurious local minima. The theory combines the features of the two mentioned works, as it applies to the entire optimization landscapes, allows for a wide spectrum of loss functions and activation functions, and constraint and unconstraint estimation. The proof defines two important concepts: path relation and block parameters and utilizes these two concepts and propositions and lemmas developed upon them to prove the main result.


The major concern of this work is that the setting is far from practice. In other words, the theory cannot provide much practical guidance nor insights for improvement.

1. First of all, the network width, $w >= 2m(n+1)^l$, is too large to be practical. Since $n$, the number of training examples, and $l$, the number of layers of a network, are large most of the times in the field today, the theorem is not very practical and not applicable in real networks.

2. Second, from the work of Zhang et al. "Understanding deep learning requires rethinking generalization”, we know that finding the global optimal solution for the training loss of overparameterized networks does not imply good generalization.  Therefore, it is not very meaningful to find the global solutions, since the network is highly overparameterized and not every global solution generalizes the same.

3. Third, the result is far from practice in deep network training. In practice, people, in general, believe that there are spurious local minimizers in deep networks (see Itay et al. “Spurious local minima are common in two-layer relu neural networks”, “Visualizing the Loss Landscape of Neural Nets”), but the data has structures and the deep network is enforcing implicit regularizations to avoid those bad local minimizers.

---

### Official Review · AnonReviewer4 · 2020-11-01
**Official Blind Review #4**

**Rating:** 4
**Confidence:** 4

**Review:**

[Summary] This paper considers the optimization landscape of deep learning with very wide networks. The main contribution of this paper is the result that shows the empirical risk of very wide networks has no *spurious local minima*. The results hold for both constrained (like the norm or sparsity constraints on the network parameters) or unconstrained risks, but require the network width larger than $O(mn^\ell)$, where $m$ is the output dimension, $n$ is the number of training samples, and $\ell$ is the number of layers.

[Pros] This paper provides interesting results on the optimization landscape of deep learning. These results could provide a better understanding of the training problems in deep learning and why the neural networks can be successfully trained by local search algorithms like SGD.

[Cons] 1. My first major comment is about the definition of *spurious local minima* used in this paper. People usually say a local minimum as a spurious local minimum if it is not a global minimum, and hence say a function has no spurious local minima if every local minimum is global. This is different from what has been used in this paper (Definition 1), which only counts a subset of local minima and excludes other local minima that are also not global. For example, see the local minima in Figure 1.a in [A].

[A] On the Benefit of Width for Neural Networks: Disappearance of Basins (https://arxiv.org/abs/1812.11039)

2. Indeed, I feel like the definition of *spurious local minima* used in this paper is similar to the so-called *set-wise strict local minimum* in the above-mentioned reference [A]. [A] provides similar results but is missing. The authors should compare the results with [A]. In particular, [A] also proves the empirical risk of networks has no spurious *set-wise strict local minimum* in the sense that every set-wise strict local minimum contains a global minimum, but only requires the width of the last hidden layer larger than $n$.

3. The results require a huge-width network, i.e., the network width larger than $O(mn^\ell)$. In this case, it is known that the network acts as a kernel method (also known as "lazy training") and many results have shown simple gradient descent converges to a global minimum. So in this sense, the result that the training problem in this regime has a nice optimization landscape is not that surprising.

---

### Official Review · AnonReviewer5 · 2020-11-05
**An**

**Rating:** 6
**Confidence:** 3

**Review:**

Summary: The paper demonstrates a claim "No spurious local minima" in the optimization landscapes of deep neural network given the width is O(n^l), where n is the number of training samples and l is the depth.

Strong points:
1. The "No spurious local minima" claim holds for all depth and quite general activation function. This greatly extends previous result and understanding.
2. The proof is heavily based on a new concept "path equivalence" and the algebraic manipulation. It is nontrivial and different from previous techniques. I checked part of the proof, which are correct to me.

Weak points:
1. The paper is hard to read for the various calligraphic notations. Although there are some description of the proof, it is better to give more intuitive argument before going to details.
2. The result requires extremely width O(n^l), under this regime each sample is guaranteed to be separable. This requirement is even larger than the NTK regime under which the gradient descent is guaranteed to find global minima. Thus it is doubtable of the theoretical/practical value of the work: use an even larger width and describe an optimization landscape that is not related with optimization methods.

Despite of the above weakness, I will recommend its acceptance to foster diverse techniques.

---

### Decision · Program_Chairs · 2021-01-07
**Final Decision**

**Decision:**

Reject

**Comment:**

This paper studies an interesting problem: the landscape of neural networks. I agree with the authors' comment that this work improves our understanding of one aspect of neural networks, and I do find the result of this paper is of interest to some extent. Reviewer 5 pointed out the technique used in the paper is interesting, and Reviewer 3 has shown interest in the techniques (and indicated the possibility of increasing the score). Nevertheless, a few reviewers questioned the requirement of the large width; I do not think having a large width itself is necessarily an issue (even in the presence of convergence results on NTK), but it is necessary to clearly explain the context and the relation/differences with closely related works in the literature. In the current form, the paper probably has not reached the bar of acceptance, thus I recommend reject.